# Currently prescribed drugs in the UK that could upregulate or downregulate ACE2 in COVID-19 disease: a systematic review

Hajira Dambha-Miller [1,2] Ali Albasri [3] Sam Hodgson [1]
Christopher R Wilcox,[1] Shareen Khan,[4] Nazrul Islam [2,5] Paul Little [1]
Simon J Griffin[2]

¹Department of Primary Care, University of Southampton, Southampton, UK
²MRC Epidemiology Unit, University of Cambridge, Cambridge, UK
³Department of Primary Care Health Sciences, University of Oxford, Oxford, UK
⁴Oxford University Hospitals NHS Trust, Oxford, UK
⁵Department of Population Health, University of Oxford, Oxford, UK

**Correspondence to**
Dr Hajira Dambha-Miller;
hajiradambha@doctors.org.uk

## ABSTRACT

**Objective** To review evidence on routinely prescribed drugs in the UK that could upregulate or downregulate ACE2 and potentially affect COVID-19 disease.
**Design** Systematic review.
**Data source** MEDLINE, EMBASE, CINAHL, the Cochrane Library and Web of Science.
**Study selection** Any design with animal or human models examining a currently prescribed UK drug compared with a control, placebo or sham group, and reporting an effect on ACE2 level, activity or gene expression.
**Data extraction and synthesis** MEDLINE, EMBASE, CINAHL, the Cochrane Library, Web of Science and OpenGrey from inception to 1 April 2020. Methodological quality was assessed using the SYstematic Review Centre for Laboratory animal Experimentation (SYRCLE) risk-of-bias tool for animal studies and Cochrane risk-of-bias tool for human studies.
**Results** We screened 3360 titles and included 112 studies with 21 different drug classes identified as influencing ACE2 activity. Ten studies were in humans and one hundred and two were in animal models None examined ACE2 in human lungs. The most frequently examined drugs were angiotensin receptor blockers (ARBs) (n=55) and ACE inhibitors (ACE-I) (n=22). More studies reported upregulation than downregulation with ACE-I (n=22), ARBs (n=55), insulin (n=8), thiazolidinedione (n=7) aldosterone agonists (n=3), statins (n=5), oestrogens (n=5) calcium channel blockers (n=3) glucagon-like peptide 1 (GLP-1) agonists (n=2) and Non-steroidal anti-inflammatory drugs (NSAIDs) (n=2).
**Conclusions** There is an abundance of the academic literature and media reports on the potential of drugs that could attenuate or exacerbate COVID-19 disease. This is leading to trials of repurposed drugs and uncertainty among patients and clinicians concerning continuation or cessation of prescribed medications. Our review indicates that the impact of currently prescribed drugs on ACE2 has been poorly studied in vivo, particularly in human lungs where the SARS-CoV-2 virus appears to enact its pathogenic effects. We found no convincing evidence to justify starting or stopping currently prescribed drugs to influence outcomes of COVID-19 disease.

### Strengths and limitations of this study

► Human and animal models both in vivo and in vitro were included for a comprehensive review.
► At the time of submission, this was the first systematic review on UK prescribed drugs that could alter ACE2 in COVID-19 disease.
► Meta-analysis was not possible due to heterogeneity.
► Methodological quality of the studies was low overall.

## INTRODUCTION

The coronavirus SARS-CoV-2 that causes the COVID-19 disease is a global public health emergency. It has been reported in 190 countries with 4 310 786 confirmed cases and 290 455 deaths as of 12 May 2020. Walker *et al* from the World Health Organization Collaborating Centre for Infectious Disease Modelling predicted that in the absence of mitigation strategies, the virus would infect 7 billion people and account for 40 million deaths this year alone.[1] Efforts to shield the elderly (60% reduction in social contacts) and interrupt transmission (40% reduction in social contacts for the wider population) have reduced this number but further deaths are still expected.[1] There is an urgent need for solutions. In the absence of a vaccination or effective treatment, there is growing interest in repurposing existing drugs for mitigation.

In particular, drugs affecting the renin–angiotensin system (RAS) have been highlighted as potential candidates for further investigation.[2 3] This is because the SARS-CoV-2 virus uses ACE2 receptors within the RAS for entry into lung alveolar epithelial cells.[4] ACE2 has previously been shown to correlate with susceptibility to the SARS-CoV-1 virus, and the spike glycoprotein of this new virus binds to ACE2

with even higher affinity.[5][6] Theoretically, altered ACE2 activity could, therefore, lead to a greater susceptibility to SARS-CoV-2. It could also cause greater severity of the infection.[7] Previous studies suggest that dysregulation of ACE2 activity in the lungs could promote early neutrophil infiltration and subsequent uncontrolled activation of the RAS.[8] In mice models, acute lung injury was observed in response to SARS-CoV-1 spike protein, so it is plausible that similar responses will be observed with SARS-CoV-2.[9] This is particularly problematic in organs containing high ACE2 such as the lungs as it may contribute to cytokine release syndrome (cytokine storm) and the subsequent respiratory failure that has been observed in those who have died from the disease.[7] Many prescribed drugs in common use are known to mediate effects through the RAS pathway. Over 45 million of these prescriptions were issued in the UK last year alone, and of these, 15 million were for ACE inhibitors (ACE-I) and angiotensin receptor blockers (ARBs). Acting through the RAS pathway, these drugs may impact ACE2 regulation but their role in the COVID-19 pandemic is not clear. Given the number of people that are potentially on these drugs, it has caused substantial public concern and clinical uncertainty about continuation or cessation of prescribed medications during the pandemic. Accordingly, we reviewed all existing evidence on routinely prescribed UK drugs that might alter ACE2 regulation. Understanding the drug effects on ACE2 given its role in COVID-19 disease could help reassure clinicians and the public in these uncertain times, or direct research on drugs that might attenuate or exacerbate transmission.

## METHODS

Our review was conducted in accordance with Preferred Reporting Items for Systematic Reviews and Meta-Analyses (PRISMA) guidelines and our protocol was submitted for open-access publication before commencing our study.[10]

### Search strategy

A systematic search in MEDLINE, EMBASE, CINAHL, the Cochrane Library and Web of Science was conducted from inception to the 1 April 2020. The full search strategy for all databases is shown in online supplemental material 1. The reference lists of recent reviews and included studies were screened. We also spoke to topic experts and screened OpenGrey for additional texts. No language limits or study design filters were applied.

### Study selection, inclusion and exclusion criteria

The COVID-19 disease is still relatively new and there is limited research on drug therapies specific to the virus. In the interest of being comprehensive about potential drugs acting through ACE2, we were as inclusive as possible within our study selection. We included both animal and human models (in vivo and in vitro). Studies had to meet the following eligibility criteria: (1) measures ACE2 levels, activity or gene expression, (2) includes a drug that is currently available on a UK prescription according to the British National Formulary and (3) measures the effect of that drug against a placebo, control or sham group in an experimental design. Review articles were excluded but their reference lists were screened. Conference abstracts were included if sufficient detail could be elicited. We did not include studies in children under 18 years, or those examining drug effects in utero.

### Data extraction

Four members of the team reviewed titles and abstracts for eligibility (AA, HD-M, CRW and SH). Full-text review, data extraction and quality assessment were carried out in duplicate using a piloted sheet. Any disagreement between authors was resolved by discussion. Data on the following study characteristics were extracted: (1) drug class, (2) drug name, (3) duration of treatment, (4) effect on ACE2 level (upregulation, downregulation and no effect), (5) model (human/rat), (6) site of ACE2 reception (lung, renal and cardiac), (7) study design, (8) study population, (9) sample size and (10) country. Given the urgency of our research question during the current pandemic, we extracted information from only what was available to us in the published text.

### Quality assessment

Our review includes both animal and human models, therefore, quality assessment was carried out separately for these studies. Human studies were evaluated using the Cochrane risk-of-bias tool, which includes the following domains: random sequence generation, allocation concealment, blinding of participants and personnel, blinding of outcome assessment, incomplete outcome data, selective reporting and other sources of bias.[11] Each domain was scored as low risk, unclear risk or high risk of bias. We classified the overall risk of bias as low if all domains were at low risk of bias, as high if at least one domain was at high risk of bias or as unclear if at least one domain was at unclear and no domain was at high risk of bias. Although this tool is specific to trials, all included studies were experimental designs and we, therefore, felt it appropriate to use. The methodological quality of animal studies was assessed using the SYstematic Review Centre for Laboratory animal Experimentation (SYRCLE) risk-of-bias tool that is based on the Cochrane risk-of-bias tool.[12] SYRCLE's tool includes selection bias, performance bias, detection bias, attrition bias, reporting bias and other biases.

### Data analysis

Owing to the mix of study designs and models, a meta-analysis was not appropriate. Narrative synthesis methods were used. We reviewed the metadata by tabulating the studies according to our inclusion/exclusion criteria, human/animal model, drug classes and effects on ACE2. The consistency in the number of studies and direction of any effects were considered. Where inconsistencies were identified in the effect of a drug between studies,

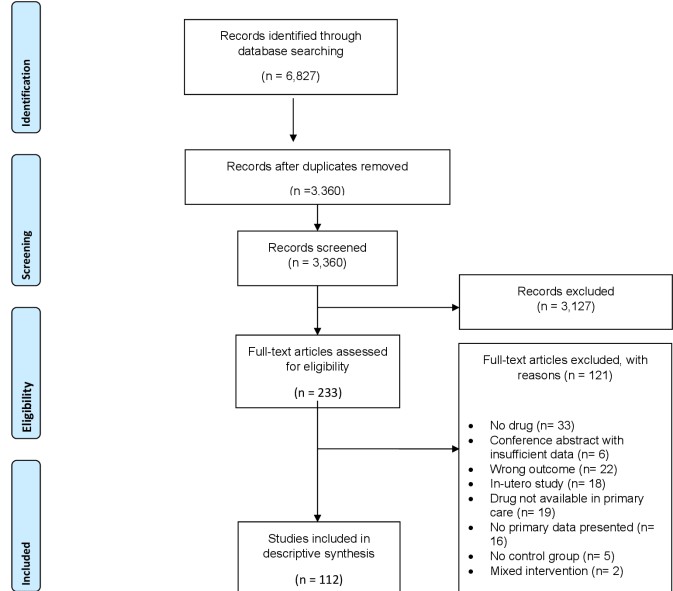

**Figure 1** Preferred Reporting Items for Systematic Reviews and Meta-Analyses flow chart explaining the study inclusion process.

we looked at additional data such as methods, quality and outcome measurement for potential explanatory factors.

### Patient and public involvement (PPI)

It was not possible to involve patients or the public in the design or conduct of our work due to the rapid timelines, but we have invited PPI representatives to help us with drafting a lay summary and in the dissemination of our findings.

### RESULTS

We retrieved 6827 studies and screened 3360 after removing duplicates. Following title and abstract screening, 233 studies were screened by full text. We included 112 studies in the final review. The flow of studies is shown in a PRISMA diagram in figure 1 including the reasons for study exclusion at each stage. A baseline table of included studies is included in the online supplemental material 2.

### Study characteristics

Table 1 shows the characteristics of the included studies. These originated from 17 different countries with the most common being China (n=36), the USA (n=22) and Japan (n=18). There were 10 studies in humans (7 in vitro and 3 in vivo) and 102 in animal models (13 in vitro and 89 in vivo). Animal models included rats (n=94), mice (n=7) and canines (n=1). The sample sizes for in vivo animal models ranged from 6 to 117. For in vivo human studies (table 2), sample sizes ranged from 8 to 375 but were not always reported. Participants were predominantly male and white with hypertension or diabetes, although the condition was not always stated. Most models examined ACE2 receptors in the heart or kidneys, only 5 of the 112

included studies reporting ACE2 levels in the lungs; these were all in animal models.

### Effects of drugs

There were 21 different drug classes examined in the included studies. Table 2 tabulates only those that have been examined in human models. The mean drug exposure period ranged from 30 min for in vitro studies to 15 weeks for in vivo studies. The most common drug classes were ARBs (n=55) and ACE-I (n=22). Of the 55 studies that examined ARBs, 43 reported upregulation of ACE2 levels. Most of these studies were in rat models (n=34) and examined cardiac ACE2 levels (n=27). For ACE-I, 17 out of 22 studies reported upregulation of ACE2. These were also mainly in rat models (n=16) and measured cardiac ACE2 levels (n=14). Of the five studies that assessed statins, these were all within rat models; three reported upregulation of ACE2, one reported downregulation and one reported no effect. Similarly, oestrogens were examined in five studies; three reported upregulation, one reported downregulation and one reported no effects. For calcium channel blockers; two out of the three studies reported upregulation of ACE2 levels and these were both in vivo rodent models. The third study was an in vitro human model that showed downregulation of ACE2 with a calcium channel blocker. There were three studies on aldosterone antagonists; all reporting increases in renal ACE2 levels within rat models.

Several diabetes drugs were evaluated and found to increase ACE2. For insulin, six out of eight studies reported upregulation of ACE2 (in mice and rat models). For thiazolidinediones, five out of seven studies reported upregulation (six mice/rate models and one of cerebral human cells in vitro). For glucagon-like peptide 1 (GLP-1) agonists, both included studies reported increases in ACE2. Similarly, for the one study examining DPP4 (Gliptans) inhibitors, it also reported an increase in ACE2. The only study measuring the effect of SGLT2 (Gliflozins) inhibitors reported a decrease in ACE2.

### Quality assessment

The risk of bias across the studies has been shown in online supplemental material 3 using the SYRCLE's risk-of-bias tool. In general, studies lacked blind allocation and outcome assessor blinding. They also frequently omitted information needed to make a thorough judgement on the risk of bias.

### DISCUSSION

To our knowledge, this is the most comprehensive review of drugs prescribed in the UK that could act on ACE2 receptors and thus potentially affect COVID-19 disease. The ACE2 receptor is reported to be an essential contributor to SARS-CoV-2 entry into the nasopharynx and lungs and the subsequent inflammation that leads to severe acute respiratory distress syndrome.[13 14] Our review examined drugs across human and animal

**Table 1** Characteristics and key findings of included studies

| Drug class | Sample size, median (range) | Exposure to treatment, mean (SD) | Effect on ACE2 expression, levels or activity (number of studies)* | Model tested (number of studies)* | Site of ACE2 receptor measurement (number of studies)* | Condition of subjects (number of studies)* |
|---|---|---|---|---|---|---|
| ACE inhibitors[27–36] | 32 (7–375) | 4 weeks (3) | Increase (n=17) Decrease (n=1) No effect (n=4) | Rats, in vivo (n=16) Humans, in vivo (n=2) Other (n=4) | Cardiac (n=14) Hepatic (n=2) Renal (n=2) Not stated (n=2) Other (n=3) | Heart disease/heart failure (n=7) Hypertension (n=3) Diabetes (n=3) Healthy (n=2) Other (n=1) Not stated (n=6) |
| Aldosterone antagonists[37–39] | 63 (28–75) | 4 weeks (4) | Increase (n=3) Decrease (n=0) No effect (n=0) | Rats, in vivo (n=3) | Renal (n=3) | Diabetes (n=1) Renal disease (n=1) Hepatic dysfunction (n=1) |
| Angiotensin receptor blockers[28 30 35 36 40–89] | 36 (6–180) | 6 weeks (6) | Increase (n=43) Decrease (n=7) No effect (n=3) Unclear/mixed findings (n=2) | Rats, in vivo (n=34) Mice, in vivo (n=11) Humans, in vivo (n=4) Other (n=6) | Cardiac (n=27) Hepatic (n=2) Renal (n=12) Lung (n=2) Not stated (n=1) Other (n=7) | Hypertension (n=13) Heart disease/heart failure (n=10) Diabetes (n=5) Healthy (n=6) Not stated (n=13) Other (n=5) |
| Beta blockers[63 90] | 52 (44–62) | 4 weeks (2) | Increase (n=0) Decrease (n=0) No effect (n=2) | Rats, in vivo (n=1) Rats, in vitro (n=1) | Cardiac (n=2) | Hypertension (n=2) |
| Calcium channel blockers[51 91 92] | 117 (N/A) | 3 weeks (1) | Increase (n=2) Decrease (n=1) No effect (n=0) | Rats, in vivo (n=2) Human, in vitro (n=1 | Hepatic (n=1) Cardiac (n=1) Cerebral (n=1) | Healthy (n=2) Hypertension (n=1) |
| Centrally acting vasodilators[86] | 6 (N/A) | 8 weeks (N/A) | Increase (n=0) Decrease (n=0) No effect (n=1) | Mice, in vitro (n=1) | Cardiac (n=1) | Other (n=1) |
| DPP4 inhibitor[93] | 24 (N/A) | 4 weeks (N/A) | Increase (n=1) Decrease (n=0) No effect (n=0) | Rats, in vivo (n=1) | Cardiac (n=1) | Healthy (n=1) |
| GABA analogues[94] | 8 (N/A) | 3 weeks (N/A) | Increase (n=0) Decrease (n=1) No effect (n=0) | Rats, in vivo (n=1) | Cerebral (n=1) | Heathy (n=1) |
| GLP-1 agonists[93 95] | 38 (24–54) | 3 weeks (1) | Increase (n=2) Decrease (n=0) No effect (n=0) | Rats, in vivo (n=2) | Cardiac (n=1) Lung (n=1) | Diabetic (n=1) Healthy (n=1) |
| Insulin[96–103] | 57 (8–84) | 6 weeks (6) | Increase (n=6) Decrease (n=1) No effect (n=1) | Mice, in vivo (n=4) Mice, in vitro (n=1) Rats, in vivo (n=2) Rats, in vitro (n=1) | Renal (n=5) Cardiac (n=2) | Diabetes (n=7) Healthy (n=1) |
| Ivabradine[104] | 24 (N/A) | 12 weeks (N/A) | Increase (n=1) Decrease (n=0) No effect (n=0) | Canine, in vivo (n=1) | Cardiac (n=1) | Heart failure (n=1) |
| NSAIDs[105 106] | 18 (N/A) | 8 weeks (0) | Increase (n=2) Decrease (n=0) No effect (n=0) | Rats, in vivo (n=2) | Cardiac (n=2) | Diabetic (n=2) |

Continued

**Table 1** Continued

| Drug class | Sample size, median (range) | Exposure to treatment, mean (SD) | Effect on ACE2 expression, levels or activity (number of studies)* | Model tested (number of studies)* | Site of ACE2 receptor measurement (number of studies)* | Condition of subjects (number of studies)* |
|---|---|---|---|---|---|---|
| Oestrogens[45 107–111] | 27 (17–75) | 3 weeks (3) | Increase (n=3) Decrease (n=1) No effect (n=1) | Human, in vitro (n=2) Human, in vivo (n=1) Rats, in vivo (n=2) Mice, in vivo (n=1) | Cardiac (n=2) Ovarian (n=1) Cerebral (n=1) Not stated (n=2) | Heart disease (n=1) Hypertension (n=1) Hypertension+ovariectomy (n=1) Healthy (n=1) Not stated (n=1) Alzheimers (n=1) |
| PDE-5 inhibitors[112] | 32 | 30 min (N/A) | Increase (n=0) Decrease (n=0) No effect (n=1) | Rats, in vitro (n=1) | Cardiac (n=1) | Healthy (n=1) |
| SGLT2 inhibitors[113] | Not stated | 15 weeks (N/A) | Increase (n=0) Decrease (n=0) No effect (n=1) | Mice, in vitro (n=1) | Renal (n=1) | Diabetic (n=1) |
| Statins[92 103 114–116] | 62 (36–87) | 5 weeks (3) | Increase (n=3) Decrease (n=1) No effect (n=1) | Rats, in vivo (n=4) Rats, in vitro (n=1) | Cardiac (n=4) Renal (n=1) | Diabetes (n=2) Hypertension (n=1) Unclear (n=1) Unclear (n=1) |
| Thiazide and thiazide-like diuretics[117] | 48 (N/A) | 1 week (N/A) | Increase (n=1) Decrease (n=1) No effect (n=0) | Rats, in vivo (n=1) | Cardiac (n=1) | Hypertension (n=1) Healthy (n=1) |
| Thiazolidinedione[105 118–123] | 21 (8–60) | 6 weeks (8) | Increase (n=5) Decrease (n=1) No effect (n=1) | Rats, in vivo (n=4) Rats, in vitro (n=1) Human, in vitro (n=1) Mice, in vivo (n=1) | Renal (n=3) Cardiac (n=2) Hepatic (n=1) Cerebral (n=1) | Hypertension (n=1) Heart disease/heart failure (n=1) Diabetes (n=1) Healthy (n=1) Renal disease (n=1) Not stated (n=2) |
| Vitamin D[123–125] | 47 (33–60) | 6 weeks (6) | Increase (n=2) Decrease (n=0) No effect (n=1) | Rats, in vivo (n=3) | Cardiac (n=1) Renal (n=1) Not stated (n=0) | Hypertension (n=1) Renal disease (n=1) Not stated (n=1) |
| Vitamin D analogues[126 127] | 28 (25–30) | 8 weeks (11) | Increase (n=1) Decrease (n=0) No effect (n=1) | Rats, in vivo (n=1) Rats, in vitro (n=1) | Renal (n=1) Lung (n=1) | Diabetes (n=5) Lung injury (n=1) |
| Zinc([128]) | Not stated | Not stated | Increase (n=0) Decrease (n=1) No effect (n=0) | Rats, in vitro (n=1) | Renal (n=1) Lung (n=1) | Not stated (n=1) |

*Studies reporting on multiple sites or in multiple models have been listed separately and appear more than once in the table.
GABA, gamma-Aminobutyric acid; GLP-1, glucagon-like peptide 1; PDE-5 inhibitor, phosphodiesterase type 5 inhibitor.

models, and we found a number of studies reporting upregulation of ACE2 levels in response to ACE-I (n=22), ARBs (n=55), insulin (n=8), thiazolidinedione (n=7) aldosterone agonists (n=3), statins (n=5), oestrogens (n=5) calcium channel blockers (n=3) GLP-1 agonists (n=2) and NSAIDs (n=2). However, these drugs were poorly studied in vivo within the lungs or nasopharynx of humans, where they are likely to matter most in influencing the severity of outcomes of COVID-19 disease.

We observed that the most frequent drugs to upregulate ACE2 are also those prescribed in people with diabetes or cardiovascular disease. Mortality rates from COVID-19 have been high in this group.[13 15–17] Notably, these are also conditions with a high prevalence among Black, Asian and Minority Ethnic groups who have had

**Table 2** Summary of studies characteristics with human models

| Drug class | Sample size, median (range) | Exposure to treatment, mean (SD) | Effect on ACE2 expression, levels or activity (number of studies) | Model tested | Site of ACE2 receptors | Condition of subject |
|---|---|---|---|---|---|---|
| Angiotensin receptor blockers | 46.5 (8–80) | 15 weeks (6) | Increase (n=2) Decrease (n=1) No effect (n=1) | In vivo (n=3) In vitro (n=1) | Urinary (n=1) Serum (n=1) Renal (n=1) Not stated (n=1) | Diabetes (n=1) Hypertension (n=1) Hypertension+diabetes (n=1) Diabetic+chronic kidney disease (n=1) |
| ACE inhibitors | 228 (80–375) | 12 weeks (N/A) | Increase (n=1) Decrease (n=1) | In vitro (n=2) | Renal (n=1) Unclear (n=1) | Diabetic+chronic kidney disease (n=1) Unclear (n=1) |
| Calcium channel blockers | N/A | Unclear | Increase ACE2 in the membrane surface (decreased in the cytosol) (n=1) | In vitro (n=1) | Cardiac | Healthy cells |
| Oestrogen | 36 (N/A) | 1 day (N/A) | Increase (n=2) No effect (n=1) | In vitro (n=3) | Cardiac (n=1) Umbilical (n=1) Not stated (n=1) | Heart problems (n=1) Healthy (n=1) Not stated (n=1) |
| Thiazolidinedione | Not stated | 1 day | Increase (n=1) | In vitro (n=1) | Cerebral (n=1) | Not stated |

disproportionally high mortality rates from COVID-19 disease.[18] To date, much of this evidence has been limited to clinical commentaries or case reports. Larger cohorts are emerging but have not yet adequately considered a range of potential confounders including comorbidities, age, sex, deprivation or household numbers which might be more important than prescribed medication in the spread, susceptibility and severity of the disease. For example, in a cohort of 191 people who were infected with the virus in Wuhan, 87% (approximately 155) of those who died had coronary heart disease and 47% (approximately 90) had diabetes.[19] These conditions are associated with an increased risk of death but were not considered as covariates in the analysis. Irrespective of ACE2, people with diabetes are more susceptible to worse infection as the low-grade chronic inflammation and hyperglycaemia associated with the condition results in impaired immune responses with lower interleukin 1 (IL-1), IL-6, tumor necrosis factor (TNF)-α and delayed mobilisation of immune cells in response to pathogens.[20] This comorbidity, like many other confounders, is highly relevant when examining the risk of death with COVID-19 disease.

This lack of adequate adjustment for existing conditions is highlighted by Sommerstein *et al* in their editorial

on ACE-I and ARBs in COVID-19.[16] They also propose that existing comorbidities such as heart failure may be independently linked to SARS-CoV-2 transmission and severity, and the subsequent poor pulmonary outcomes that are observed in these patients. Indeed, in mice models, arterial hypertension, atrial fibrillation and type 2 diabetes have been shown to upregulate ACE2 levels irrespective of medications.[21 22] Moreover, ACE2 levels are higher in men and with increasing age.[23] Most of the published data on deaths in COVID-19 disease report that men of increasing age are particularly susceptible to poor outcomes.[13 24]

Our review has also highlighted the variable ACE2 levels in different parts of the body with most of the existing literature focussing on renal and cardiac levels. Responses to drugs may vary depending on cell type and location. Although the lung ACE2 is important to COVID-19, it is unclear if overall COVID-19 mortality might be attenuated by cardiovascular ACE-2 activity levels. We also observed variations in ACE2 levels with drug exposure duration which was relatively short among included studies in our review. It is uncertain how dysregulation might continue after starting or stopping these medications. It is also unclear how the observed effects among included studies would translate in vivo in humans and

what the net effect on receptor access to the COVID-19 virus is; access to the receptor by the virus may be competitively inhibited by the presence of drugs which also attach to the receptor, so whether upregulation is the key factor in practice is unclear. This is particularly challenging to understand as we found a paucity of data demonstrating the effect of prescribed drugs on ACE2 in the lungs or nasopharynx, where the SARS-CoV-2 virus appears to enact its pathogenic effects. Our results, therefore, do not provide convincing evidence on the role of any currently prescribed UK drugs acting through ACE2 regulation that could affect COVID-19 disease. Finally, we found a disproportionate number of studies reporting upregulation or 'positive effects' of drugs on ACE2, compared with studies reporting no effect or downregulation. This may reflect a publication bias that is well established in the literature, especially among animal models.[25 26]

## Strengths and limitations

We carried out a comprehensive and systematic search of the literature. To our knowledge, at the time of submission (April 2020), this is the first review on the subject. We did not include language restrictions but non-English language studies in the international literature might not have been indexed in the databases we searched. Given the rate of new publications on COVID-19, it is also possible that our search and results may not be up to date. Owing to the limited research on this novel virus, it was necessary to be as inclusive as possible and we, therefore, considered both animal and human models to look for any drugs acting through ACE2 with potential to affect COVID-19 outcomes. While this inclusive approach may offer insights, the heterogeneity across models makes it hard to interpret findings or translate them directly to patients. We did not formally assess heterogeneity but this is likely as we had multiple different models including animal, human, in vitro, in vivo as well as different body sites (heart, lung and kidney) Future studies are needed that can quantify effects through meta-analysis, and examine dose responses. Although we were robust in our methodological approach to this review, we were also aware of the urgency to report our findings in the current pandemic. We, therefore, did not contact authors for more information about their studies beyond what was published. We observed frequent omission of information that would have allowed us to carry out a more detailed quality assessment. Had we pursued this information; the quality assessment of included papers may well have been higher. At present, all studies were high risk of bias, which is a limitation of this review.

## CONCLUSION

We reviewed the evidence on routinely prescribed drugs in the UK that could upregulate or downregulate ACE2, and thus potentially affect COVID-19 disease. Our review indicates that currently prescribed drugs have been poorly studied in vivo within the lungs of humans. Until there is better evidence, we cannot recommend starting or stopping prescribed medications during the COVID-19 pandemic.

**Acknowledgements** We would like to thank Professor Julia Hippisley-Cox for her early contributions to this project.

**Contributors** HD-M contributed to the design of the study, wrote the analysis plan, conducted the analysis, drafted and revised the paper. AA contributed to the design of the study, led the analysis, drafted and revised the paper. CRW and SH contributed to the screening of studies, data extraction and revised the paper. SK contributed to screening of studies. NI revised the paper. SJG and PL contributed to the design of the study and revised the paper. HD-M was the guarantor.

**Funding** The Southampton, Cambridge and Oxford Primary Care Departments are members of the NIHR School for Primary Care Research and supported by NIHR Research Funds. The University of Cambridge has received salary support in respect of SJG from the National Health Service (NHS) in the East of England through the Clinical Academic Reserve. SJG is supported by an MRC Epidemiology Unit programme: MC_UU_12015/4. HD-M is an NIHR Academic Clinical Lecturer. The views expressed are those of the author(s) and not necessarily those of the NHS, the NIHR or the Department of Health and Social Care

**Competing interests** None declared.

**Patient and public involvement** Patients and/or the public were involved in the design, or conduct, or reporting, or dissemination plans of this research. Refer to the Methods section for further details.

**Patient consent for publication** Not required.

**Provenance and peer review** Not commissioned; externally peer reviewed.

**Data availability statement** Data are available upon reasonable request.

**ORCID iDs**
Hajira Dambha-Miller http://orcid.org/0000-0003-0175-443X
Ali Albasri http://orcid.org/0000-0001-7805-1965
Sam Hodgson http://orcid.org/0000-0002-5610-850X
Nazrul Islam http://orcid.org/0000-0003-3982-4325
Paul Little http://orcid.org/0000-0003-3664-1873

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
