## [Reviewer comments · BMJ Open]

ARTICLE DETAILS

TITLE (PROVISIONAL)	Currently prescribed drugs in the UK that could up or downregulate ACE2 in COVID-19 disease: A systematic review
AUTHORS	Dambha-Miller, Hajira; Albasri, Ali; Hodgson, Sam; Wilcox, Christopher; Khan, Shareen; Islam, Nazrul; Little, Paul; Griffin, Simon

VERSION 1 - REVIEW

REVIEWER	Sean Zheng Imperial College London, UK
REVIEW RETURNED	05-Jun-2020

GENERAL COMMENTS	Currently prescribed drugs in the UK that could up or downregulate ACE2 in COVID-19 disease: A systematic review Reviewer comments The systematic review by Dambha-Miller and colleagues addresses an interesting and topical issue in the current unique clinical environment. The purpose of the study is to identify systematically the effect of different pharmacological agents on up and downregulation of ACE2 in in vivo and in vitro studies (human and animal models). I have no major concerns. The study is appropriately reported and follows a standard protocol for systematic reviews. Major Please report and present PRISMA checklist. Minor Table of summary results reports a total of 6827, however study flow chart and Results report 6821. Please reconcile. Heterogeneity was not assessed. Authors presumed and anticipated that heterogeneity would be high and therefore not appropriate for meta-analysis, which I agree with. This however was never formally tested in an a priori manner. The authors should not make statements pertaining to heterogeneity in the Strengths and Limitations sections of the Discussion and Brief Overview that give the impression that heterogeneity was formally assessed. Please rephrase. All studies were high risk of bias. Please report this as a limitation of study.
--

REVIEWER	Giuseppe Biondi-Zoccai Sapienza University of Rome, Latina, Italy
REVIEW RETURNED	16-Jun-2020

GENERAL COMMENTS	The authors report an interesting systematic review on drugs potentially leading to overexpression of ACE2 receptors. This is timely and potentially informative, give the involvement of these receptors in SARS-CoV-2 infection, and also possibly given that they may impact on the likelihood and severity of subsequent COVID-19 (eg Infusino et al, Relationship between ACE-inhibitors, ARBs and SARS-CoV-2 infection: where are we? Minerva Cardioangiol. 2020 May 29. doi: 10.23736/S0026-4725.20.05271-8). Despite the work strengths, I recommend addressing the following comments:  1. Methods: Double check that no issue of duplicate publication is present. Several papers from a few prolific groups are evident and dates of research activities should be double checked to minimize this risk. 2. Methods and Results: Try to disentangle between direct vs indirect effects of drugs. For instance, as ACE2 receptors are involved in blood pressure control, any drug or condition impacting on such feature (eg shock) could impact on ACE2 expression/activity (eg Yang et al, Effects of Angiotensin II Receptor Blockers and ACE (Angiotensin-Converting Enzyme) Inhibitors on Virus Infection, Inflammatory Status, and Clinical Outcomes in Patients With COVID-19 and Hypertension: A Single-Center Retrospective Study. Hypertension. 2020 Jul;76(1):51-58. doi: 10.1161/HYPERTENSIONAHA.120.15143). 3. Results: Try to conduct some form of quantification of effect. For instance, some drugs may impact more than others on ACE2. Similarly, was any dose-finding relationship found? This would strengthen your case for any given drug (eg Altshuler. Modeling of Dose-Response Relationships. Environ Health Perspect. 1981 Dec;42:23-7. doi: 10.1289/ehp.814223).. 4. Figure: Add at least one high-quality and original figure to summarize in a multidimensional fashion your findings. 5. Throughout: Check the manuscript for occasional typos (eg "ewsadxzAll" in the Competing Interests section).
--

VERSION 1 – AUTHOR RESPONSE

Reviewer(s)' Comments to Author:	
Please report and present PRISMA checklist.	A PRISMA checklist has been included
Table of summary results reports a total of 6827, however study flow chart and Results report 6821. Please reconcile.	Thank you for picking up on this – the number in the PRISMA diagram has been corrected to 6827.
Heterogeneity was not assessed. Authors presumed and anticipated that heterogeneity would be high and therefore not appropriate for meta-analysis, which I agree with. This however was never formally tested in an a priori manner. The authors should not make statements pertaining to heterogeneity in the Strengths and Limitations sections of the Discussion and Brief	We have altered the text in our strengths and limitations to make it clear that heterogeneity was not formally tested.

Overview that give the impression that heterogeneity was formally assessed. Please rephrase.	
All studies were high risk of bias. Please report this as a limitation of study.	Thank you, we have included this within the limitations.
Methods: Double check that no issue of duplicate publication is present. Several papers from a few prolific groups are evident and dates of research activities should be double checked to minimize this risk.	We have double checked for duplication. A few groups appear more than once only where authors have reported on different models and sites. The top of our table already clarifies this with a note that reads: Studies reporting on multiple sites or in multiple models have been listed separately and appear more than once in the table
Methods and Results: Try to disentangle between direct vs indirect effects of drugs. For instance, as ACE2 receptors are involved in blood pressure control, any drug or condition impacting on such feature (eg shock) could impact on ACE2 expression/activity (eg Yang et al, Effects of Angiotensin II Receptor Blockers and ACE (Angiotensin-Converting Enzyme) Inhibitors on Virus Infection, Inflammatory Status, and Clinical Outcomes in Patients With COVID-19 and Hypertension: A Single-Center Retrospective Study. Hypertension. 2020 Jul;76(1):51-58. doi: 10.1161/HYPERTENSIONAHA.120.15143).	We agree this is an important question that needs further investigation. Our data and included studies did not supply sufficient detail to determine whether effects of drugs were direct or indirect. This is particularly problematic in cell models and animal models. However, this is certainly a question that needs investigation is future research.
Results: Try to conduct some form of quantification of effect. For instance, some drugs may impact more than others on ACE2. Similarly, was any dose-finding relationship found? This would strengthen your case for any given drug (eg Altshuler. Modeling of Dose-Response Relationships. Environ Health Perspect. 1981 Dec;42:23-7. doi: 10.1289/ehp.814223)..	We thank the reviewer for these insightful comments and they are correct about the need for quantification and dose-response. As explained in our manuscript, quantification was not possible with the data that is available within included studies especially given the number of different study designs and models across cell lines and in vivo animal models. Each study used different drugs rather than testing variations of the same drug in the same model to allow dose response assessment. We agree 100% that this need standardisation and exploration in future studies has been emphasised to permit quantification
Figure: Add at least one high-quality and original figure to summarize in a multidimensional fashion your findings.	We have provided a higher-quality file for our key findings (table 1 and table 2)
Throughout: Check the manuscript for occasional typos (eg "ewsadxzAll" in the Competing Interests section).	We apologise and have corrected this wherever they appear.

VERSION 2 – REVIEW

REVIEWER	Giuseppe Biondi-Zoccai Sapienza University of Rome, Latina, Italy
REVIEW RETURNED	30-Jul-2020

GENERAL COMMENTS	All my comments have been reasonably addressed.
---